# Pragmatic, quasi-experimental, pseudo-randomized clinical trial to assess the impact of patient safety monitors on clinical and patient safety outcomes: The Akershus Clinical Trial (ACT) 1

Kristian Berge[1,2,3], Torbjørn Wisløff[2,4], Kristine Lippestad[2,5], Rune Bruhn Jakobsen[6], Johanna A. Gjestland[6], Olav Lenvik[5], Haldor Husby[2,5], Magnus N. Lyngbakken[1,2,3], Inge Skråmm[6], Helge Røsjø[2,3]*

1 Division of Medicine, Akershus University Hospital, Lørenskog, Norway, 2 Akershus Clinical Research Center (ACR), Division of Research and Innovation, Akershus University Hospital, Lørenskog, Norway, 3 K. G. Jebsen Centre for Cardiac Biomarkers, Institute of Clinical Medicine, University of Oslo, Oslo, Norway, 4 Department of Health Services Research, Division of Research and Innovation, Akershus University Hospital, Lørenskog, Norway, 5 Department of Analytics, Division of Economy and Finance, Lørenskog, Norway, 6 Department of Orthopedic Surgery, Akershus University Hospital, Lørenskog, Norway

* helge.rosjo@medisin.uio.no

## Abstract

### Background

The Norwegian National Action Plan for Patient Safety and Quality Improvement recommends implementing electronic patient safety monitors (PSM) in Norwegian hospitals. We hypothesized that PSM implementation would reduce hospital length of stay, with secondary endpoints related to clinical and patient safety outcomes.

### Methods

We performed a retrospective single-center, pragmatic, quasi-experimental, pseudo-randomized controlled trial assessing differences in endpoints for patients admitted before (control group: years 2019–2020) and after PSM implementation (intervention group: years 2021–2022) in two orthopedic bays in a Norwegian teaching hospital. To account for temporal changes unrelated to PSM, we assessed the same endpoints for 2019–2020 vs. 2021–2022 in two orthopedic bays that did not implement PSM. Data extraction from the local data warehouse was controlled according to internal standards, and we predefined all statistical analyses.

### Results

The intervention group (n = 2786) and the control group (n = 2610) were evenly distributed for baseline characteristics, including age (median 73 [quartile 1–3 55–83]

which permits unrestricted use, distribution, and reproduction in any medium, provided the original author and source are credited.

**Data availability statement:** The dataset underlying this study contains sensitive patient information and cannot be made publicly available, as the approval from the Norwegian Data Protection Authority and patient consent do not permit open sharing of individual-level data. De-identified data may be made available upon reasonable request for researchers who meet the criteria for access to confidential data. Data requests can be directed to the Akershus University Hospital Clinical Research Unit (ACR), which serves as the long-term institutional data custodian (contact: acr@ahus.no).

**Funding:** The author(s) received no specific funding for this work.

**Competing interests:** KB has received speaker honoraria from Boehringer Ingelheim, Novartis, Amgen, and AstraZeneca.

vs. 72 [55–83] years, p = 0.29), female sex (76.2% vs. 76.4%, p = 0.96), and Charlson Comorbidity Index (p = 0.24). Median hospital length of stay was 4.4 (2.1–7.7) days after implementation of PSM compared to 4.3 (2.0–7.2) days in the control group (p = 0.046 for difference between the groups). Nutritional screening and fall screening within 24 hours were lower in the intervention group compared to the control group, while medical reconciliation during the hospital stay was higher in the intervention group. We found no differences between the groups in 30-day or 1-year readmission or mortality rates. Hospital length of stay was similar in 2021–2022 vs. 2019–2020 for the bays that did not implement PSM.

## Conclusion

The implementation of PSMs in this study did not improve clinical or patient safety outcomes. Hospital length of stay was statistically longer after PSM implementation, although the difference was not clinically relevant.

## Introduction

Patient safety remains a significant concern in modern healthcare systems. Recent data indicate that 12% of patients experience healthcare-related harm, of which half of the cases are preventable [1]. Among the preventable cases of healthcare-related patient harm, 12% will lead to severe morbidity or death [1]. This makes patient harm the 14th leading cause of global disease burden, which is comparable to the burden from HIV/AIDS and traffic accidents. Preventable healthcare-related harm accounts for 15% of total hospital expenditure [2] and 4% of all hospital deaths [3]. To reduce avoidable harm and improve patient safety, the World Health Organization (WHO) has developed a Global Patient Safety Action Plan [4], which provides a framework for countries to develop their respective national action plans on patient safety.

In Norway, efforts to improve patient safety are grounded in the National Action Plan for Patient Safety and Quality Improvement (2019–2023) [5]. This strategic initiative targets key areas like reducing healthcare-associated infections, minimizing medication errors, and enhancing coordinated care for patients with complex conditions. The plan emphasizes the importance of a systematic safety approach, integrating new technologies, and better communication strategies. For in-hospital settings, one of the recommendations of the action plan is to implement brief multidisciplinary meetings around huddle boards. During these meetings, there is a focus on routines to prevent pressure ulcers and falls and to identify patients at high risk of adverse events. However, limited information is available on the clinical value of huddle boards. In addition, electronic health records now provide an opportunity to integrate data for individual patients and report these back in structured format to clinical personnel on large screens, so-called Patient Safety Monitors (PSMs). Currently, there is limited information in the literature on whether PSM improves clinical and patient safety outcomes. Accordingly, in this study, we hypothesized that the implementation

of PSMs in two orthopedic bays in a Norwegian teaching hospital would reduce hospital length of stay (LOS) (primary outcome) and improve clinical and patient safety outcomes (secondary outcomes).

## Methods

### Study design

We conducted a pragmatic, quasi-experimental, pseudo-randomized clinical trial using routinely collected data from the electronic health records (EHR) at Akershus University Hospital. A predefined study protocol and statistical analysis plan were approved by the study group and electronically time-stamped prior to study initiation (S1 and S2 Appendices). Patients were assigned to intervention or control groups based on the timing of hospital admission, rather than by true random allocation. Specifically, patients admitted to two orthopedic bays during the two years preceding the implementation of PSMs constituted the control group, while those admitted in the subsequent two years following implementation formed the intervention group. Although allocation was calendar-based rather than truly random, this approach functions as a form of pseudo-randomization, since patient assignment was independent of individual characteristics and was therefore expected to yield balanced baseline characteristics between groups. This natural allocation allows for evaluation of PSM effectiveness under real-world conditions, closely reflecting routine clinical practice.

### Patient population and PSM implementation in the orthopedic ward

The study population consists of all patients admitted to two orthopedic bays (multi-bed subunits of the ward, typically hosting 6–10 patients each) at Akershus University Hospital between 01/01/2019 and 31/12/2022 who implemented PSM (DNV Imatris, Porsgrunn, Norway) starting 01/01/2021. Akershus University Hospital is a Norwegian teaching hospital with a catchment area of more than 600,000 people and has the largest orthopedic department in Norway. PSMs were introduced in the two orthopedic bays in August 2020, with the initial months designated as a pilot and introduction period. An example screenshot of the PSM user interface is provided in Fig 1. The PSM replaced the established routine, where patient safety data was accessible as a dashboard on the healthcare workers' computers. Orthopedic patients were considered a relevant population for the initial implementation of PSMs, as this group is at increased risk of preventable adverse events such as pressure ulcers, venous thromboembolism, delirium, and falls, making patient safety monitoring particularly pertinent in this setting. Patients aged 18 years or older were eligible for inclusion in the study.

The PSM at Akershus University Hospital during the study period provided individual real-time data for selected clinical variables of all patients in the designated bays, including all variables used as secondary outcomes of the study (i.e.,

**Fig 1. Example screenshot of the Patient Safety Monitor (PSM) interface.** The electronic board displays patient-level information relevant to safety processes in real time. Columns include: patient ID (anonymized here), age, admission date, NEWS2 early warning score, peripheral venous catheter (PVC) status, medication reconciliation, physician contact time, prophylaxis (e.g., anticoagulation, antibiotics, osteoporosis therapy), allied health services (physiotherapy, social/municipal health services), discharge planning, nutritional screening, pressure ulcer screening, fall risk screening, and delirium screening (4AT). Color coding highlights overdue tasks (red), completed items (green), and intermediate status (orange). This screenshot illustrates the system in routine use on an orthopedic ward.

National Early Warning Score 2 [NEWS2] scoring, nutritional screening, fall screening, administration of anti-coagulation, and medical reconciliation). The PSM receives this information from several electronic health systems, which are integrated by the Department of Analytics using the data warehouse of Akershus University Hospital, and the information is presented on the digital monitors with an update every 15 minutes. The PSM will signal with the change of color if predetermined patient safety parameters are not met (e.g., red color if a safety parameter has not been executed according to protocol). The PSM was placed on the station´s wall for nurses in the bays, where it served as a visual reminder of patient safety parameters throughout the workday.

Patients hospitalized in these bays during the two years before PSM implementation (01/01/2019–31/12/2020) were defined as the historical control group, while patients hospitalized in the following two years (01/01/2021–31/12/2022) were defined as the intervention group. These bays primarily accommodate acute orthopedic patients, and the patient groups along with the clinical services provided in these two bays remained unchanged from 2019 to 2022. We present data exclusively for unique patients. Specifically, if a patient was hospitalized multiple times during the study period, only the initial hospitalization is considered the index hospitalization, while any subsequent hospitalizations are regarded as clinical events.

To account for temporal variations unrelated to PSM, data were also acquired from the years 2019–2020 and 2021–2022 from two additional orthopedic bays within Akershus University Hospital where PSM was not implemented.

### Individual patient data

The Norwegian Directorate of Health granted an exception from confidentiality requirements, allowing the collection of individual patient data without requiring written or verbal consent. Data were extracted on 25/10/2023 from the HER systems at Akershus University Hospital, using queries to the hospital's centralized data warehouse. The source data were originally entered into various EHR systems by clinical staff during routine patient care.

### Data collection for outcomes and follow-up

All endpoints were predefined in the study protocol and the statistical analysis plan (S1 and S2 Appendices). The primary outcome of the study was a change in hospital LOS compared to the historical control group. Data on hospital LOS were extracted from the EHR system (Distributed Information and Patient Data System in Hospitals, DIPS, Bodø, Norway). All ward accommodations within the Department of Orthopedics were included in the hospital stay. The dataset was complemented with subsequent hospital stays in the study bays, aiming to investigate whether the 30-day and 1-year readmission rate was changed compared to historical control patients, which were secondary endpoints of the study. Furthermore, time of death was collected to determine the change in 30-day and 1-year all-cause mortality. None of the authors had access to information that could identify individual participants during or after data collection.

Patient safety outcomes were obtained from the EHR system Metavision (iMDsoft, Tel Aviv, Israel). The endpoints included nutritional screening, fall screening, anti-coagulation therapy, medication reconciliation, and total number of and time to first screening of vital parameters measured by the clinical scoring system National Early Warning Score 2 (NEWS2). NEWS2 provides a score based on clinical variables like respiratory rate, oxygen saturation, hypercapnic respiratory failure, need for supplemental oxygen, systolic blood pressure, heart rate, consciousness and body temperature [6].

The covariates considered in the analyses were age, sex, Charlson Comorbidity Index, and cause and time of index admission, all harvested from the EHR system, due to their potential effect on hospital LOS and mortality. The Charlson Comorbidity Index was scored based on ICD-10 codes registered at prior hospital admissions at Akershus University Hospital. In total, 19 conditions were weighted and added up, resulting in a score on a 0–33 scale based on a previously developed algorithm [7,8].

## Data handling

A comprehensive internal quality monitoring process was conducted to verify the accuracy of the data extracted and collected from the EHR. We used an internal monitoring protocol that required at least 30 random samples to be selected for manual check-up. Accordingly, data managers manually reviewed ≥30 random samples and checked the accuracy of the automated extracted data compared to the data extracted by manual extraction. In addition, a clinical researcher also reviewed the clinical information extracted for accuracy of diagnosis and comorbidities versus the information obtained by automated data extraction.

Before and after extraction, the data manager performing the extraction and the statistician who performed statistical analyses had collaborative meetings to ensure a common understanding of all variables. Since data were extracted from different EHR systems, the data manager and statistician collaborated to create a detailed plan for extracting data, including the data source from which individual data would be extracted. The data manager and the statistician decided together how to build the database prior to the statistical analyses. The authors did not have access to the database, and all statistical analyses were performed by a dedicated statistician. The signed protocol for internal monitoring of data extraction in the study is presented in the S3 Appendix.

## Statistics

For the primary outcome, we performed the analysis as predefined in the statistical analysis plan with a generalized linear regression model with gamma family and loglink which adjusted for age, sex, Charlson Comorbidity Score, cause of index admission, and time of hospital admission. Additionally, time-to-event analyses were performed using Cox proportional hazards regression, adjusting for the same variables as the primary outcome. Analyses were performed for the first 30 days and one year, and for mortality and readmission separately. In analyses for readmission, mortality was regarded as censoring. We used logistic regression to assess changes in nutritional screening, both for 24 hours and the index hospitalization, and fall screening within 24 hours from hospital admission. Differences in a total number of NEWS2 measurements taken during hospitalization were assessed using negative binomial regression.

To assess if changes in endpoints were attributable to the implementation of PSM, we performed sensitivity analysis that examined the same outcomes in orthopedic bays where PSMs were not implemented. The effect of PSM implementation was tested by including and assessing the significance of an interaction term between the periods (pre- and post-implementation) and bay type (intervention bays vs. non-intervention bays) in the models (difference-in-difference analyses). Sample size calculations are not applicable for this study as we use a retrospective design. However, we have calculated the power of the data to demonstrate a potential change in the primary endpoint. Data from the Norwegian Patient Registry indicates that the mean length of stay in orthopedic wards is 4 days (standard deviation 2.5 days). We define a clinically relevant reduction in hospital length of stay to be at least 1 day (24 hours). Based on these numbers and expecting 1:1 ratio between hospitalized patients during the control period (2019–2020) and implementation period (2021–2022), a minimum of 100 patients in the intervention arm and 100 patients in the in silico-control arm were needed to have >80% probability to detect a difference with significance level 0.05. All analyses were predefined in a *Statistical Analysis Plan* before the initiation of this study (S2 Appendix). Regression analyses were performed in RStudio (version 2023.03.1)/ R (version 4.2.3), using the coxph command for Cox regression, glm.nb for negative binomial regression, and the glm command for all other types of regression.

## Results

### Baseline characteristics

A total of 5,396 unique patient hospitalizations from 2019 to 2022 were included in the main analysis, with 2,610 patients admitted before implementation of PSM (control period, year 2019–2020) and 2,786 patients after implementation of PSM

(intervention period, year 2021–2022) (Fig 2). The median age was 73 years (Q1–3 55–83), with 76.3% being female, and 80.0% of hospitalizations were attributed to acute medical conditions. The distribution of baseline characteristics, including age, sex, body mass index, prevalence of emergency hospitalizations, Charlson Comorbidity Index, and individual comorbidities, was not significantly different between the patients hospitalized in the intervention and control period (Table 1). We neither observed differences between patients hospitalized during 2019–2020 and those in 2021–2022 in the two orthopedic bays that did not implement PSM during the latter (intervention) period (S1 Table). Compared with the

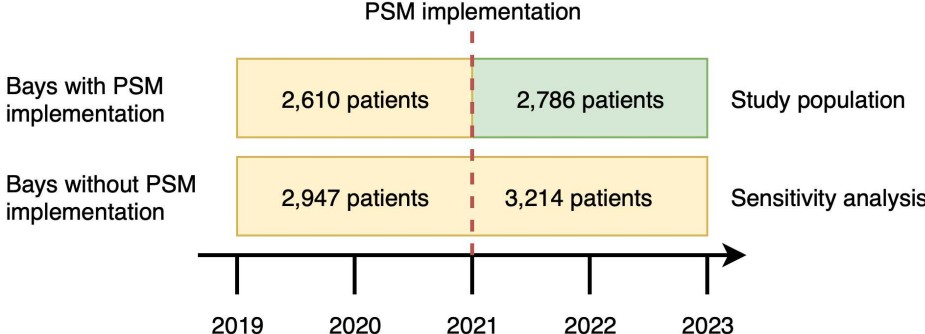

**Fig 2. Study population and sensitivity analysis.** Overview of patient allocation before and after implementation of the Patient Safety Monitor (PSM). The main study population consisted of 2,610 patients admitted prior to PSM implementation (2019–2020) and 2,786 patients admitted after implementation (2021–2022) in wards where PSM was introduced. For sensitivity analyses, 2,947 and 3,214 patients admitted during the same periods in wards without PSM implementation were included.

**Table 1. Baseline characteristics.**

|  | Intervention (n = 2786) | Control (n = 2610) | p-value* |
|---|---|---|---|
| Age, years | 73 [55-83] | 72 [55-83] | 0.29 |
| Female sex, n (%) | 2124 (76.2) | 1994 (76.4) | 0.96 |
| Charlson score, n (%) |  |  | 0.24 |
| 0 | 1371 (49.2) | 1341 (51.4) | 0.36 |
| 1 | 437 (15.7) | 424 (16.2) | 0.63 |
| 2 | 393 (14.1) | 325 (12.5) | 0.12 |
| 3 | 228 (8.2) | 179 (6.9) | 0.09 |
| ≥4 | 357 (12.8) | 341 (13.1) | 0.81 |
| Emergency hospitalization, n (%) | 2235 (80.2) | 2081 (79.7) | 0.88 |
| Medical history, n (%) |  |  |  |
| Cerebrovascular disease | 366 (13.1) | 359 (13.8) | 0.56 |
| Heart failure | 284 (10.2) | 242 (9.3) | 0.30 |
| Myocardial infarction | 258 (9.3) | 224 (8.6) | 0.43 |
| Diabetes mellitus | 325 (11.7) | 276 (10.6) | 0.26 |
| Chronic pulmonary disease | 330 (11.8) | 295 (11.3) | 0.58 |
| Malignancy | 406 (14.6) | 348 (13.3) | 0.25 |
| Dementia | 162 (5.8) | 152 (5.8) | 0.99 |
| Renal disease | 156 (5.6) | 153 (5.9) | 0.70 |
| Peripheral vascular disease | 258 (9.3) | 207 (7.9) | 0.11 |

*p-value for difference between intervention and control group, t-test for age, Chi-square test otherwise.

pre-implementation period (2019–2020), elective surgical activity was 22% lower in the intervention period (2021–2022), while emergency surgical activity remained stable (1%), and ward-level orthopedic nursing staffing increased by 18%.

## Hospital length of stay

Before the implementation of PSM, the median LOS was 4.3 [2.0–7.2] days. After the implementation of PSM, LOS had increased to 4.4 [2.1–7.7] days (p = 0.046 after adjustment for age, sex, Charlson comorbidity score, and cause of index admission; Table 2). In sensitivity analysis, examining bays that did not implement PSM revealed a trend toward longer LOS in the intervention period (3.8 [1.7–7.0] days vs. 3.3 [1.5–6.7]), although this difference was not statistically significant (p = 0.25). We did not find a significant interaction between the hospitalization period (before vs. after PSM implementation) and bay placement (bay with vs. without PSM implementation) on hospital LOS (p-for-interaction = 0.34).

## Patient safety measures, hospital readmissions, and mortality

Prevalence of nutritional screening within 24 hours and during the entire hospital stay was lower in the intervention period compared to the control period (6.0% vs. 8.6%, p < 0.001, and 10.3% vs. 14.1%, p < 0.001, respectively) (Table 2). Screening for fall risk within 24 hours was lower during the intervention period compared to the control period (7.5% vs. 10.8%, p < 0.001). We did not find a significant difference in the prescription of pharmacological thromboprophylaxis (dalteparin) between the control period (59.6%) and intervention period (57.7%, p = 0.90). NEWS2 scoring within 24 hours was not different after implementation of PSM (96.5% vs. 96.1%, p = 0.62), but the total number of NEWS2 scores during hospitalization increased after PSM implementation (12 [5–22] vs. 12 [5 –20 ], p = 0.001). The prevalence of medication reconciliation was higher after the implementation of PSM (8.4% vs. 0.6%, p < 0.001), and the same increase was also found in bays that did not implement PSM. We did not find any significant interactions between the introduction of PSM and the pre-defined safety endpoints (S2 and S3 Tables).

There were no significant differences in 30-day (6.5% vs. 7.4%, p = 0.20) or 1-year (14.0% vs. 12.9%, p = 0.27) readmission rates for patients admitted before or after the implementation of PSMs (Table 3). We found no differences in 30-day (2.9% vs. 2.8%, p = 0.52) or 1-year (11.0% vs. 11.3%, p = 0.75) all-cause mortality rates. We found similar results for bays where PSM were not implemented, with no significant differences in rehospitalization or mortality rates in the two time periods studied (S4 Table). There were no statistical interactions between the intervention period and intervention bays on readmission or mortality rates.

Table 2. Primary and secondary endpoints.

|  | Intervention (n = 2786) | Control (n = 2610) | p-value* |
|---|---|---|---|
| Primary endpoint |  |  |  |
| Hospital length of stay, days | 4.4 (2.1–7.7) | 4.3 (2.0–7.2) | 0.046 |
| Secondary endpoints, n (%) |  |  |  |
| Nutritional screening within 24 hours | 168 (6.0%) | 224 (8.6%) | <0.001 |
| Nutritional screening during hospitalization | 288 (10.3%) | 369 (14.1%) | <0.001 |
| Fall screening within 24 hours | 208 (7.5%) | 282 (10.8%) | <0.001 |
| Pharmacologic thromboprophylaxis | 1608 (57.7%) | 1556 (59.6%) | 0.072 |
| NEWS2 scoring within 24 hours | 2688 (96.5%) | 2509 (96.1%) | 0.62 |
| Number of NEWS2 scores during hospitalization | 12 (5–22) | 12 (5–20) | 0.001 |
| Medication reconciliation | 234 (8.4%) | 16 (0.6%) | <0.001 |

*All analyses adjusted for age, sex, Charlson comorbidity score, cause of index admission, and admission period (2019–2020 vs. 2021–2022).

## Discussion

The principal finding of our study was that implementation of electronic PSMs was associated with a statistically significant but clinically negligible increase in hospital LOS. Implementation of PSMs did not improve several clinical and patient safety outcomes for patients hospitalized in the orthopedic bays of a large Norwegian teaching hospital. Additionally, there was a decrease in the proportion of patients receiving nutritional and fall screenings within 24 hours. Medication reconciliation increased during the period studied, both in bays that implemented PSM and bays that did not implement PSM.

Hospital safety measures are critical to reduce healthcare-associated harm and improve patient outcomes. Among the most significant advancements in patient safety was the introduction of the WHO Surgical Safety Checklist, which reduced surgical complications by 36% and mortality by 47% in a prospective multicenter trial [9]. Similarly, the use of multidisciplinary team huddles has demonstrated improvements in communication, teamwork, and situational awareness, collectively reducing the risk of patient harm [10,11]. A recent systematic review showed that huddles significantly reduce patient falls, hospital LOS, and readmissions [11]. Further building on this, the WHO's *Global Patient Safety Action Plan* highlighted that adapting digital real-time monitoring of health information technologies such as digital PSMs could further improve patient safety [4]. While high-quality studies on the implementation of PSM do not exist, a systematic review of digital dashboards for the visual display of patient safety data demonstrated the value of such measures, particularly in detecting missed doses of medications, and reducing rates of hospital-acquired infections and pressure ulcers [12]. Still, the systematic review highlights that the level of evidence in the research field is limited, with most studies reporting results from quality improvement processes containing multiple interventions, which limits the possibility to extract specific information on PSM implementation from these studies.

In our study, we did not observe improved outcomes related to hospital LOS or clinical endpoints and patient safety measures that could be attributed to PSM implementation. Despite international endorsement of digital patient safety initiatives, numerous quality improvement intervention studies have demonstrated neutral or negative results. A recent scoping review assessed the effect of implementation of electronic systems on patient safety in interventional studies, with most studies assessing the implementation of electronic notification systems [13]. Among 14 predefined safety endpoints, they only found strong evidence for one intervention on patient safety: the reduction of adverse drug events due to increased medication reconciliation. In contrast, the scoping review reported no or limited effect by introducing electronic systems on the prevalence of infections, venous thromboembolism, pressure ulcers, mechanical complications and underfeeding, clinical pathway, and safety culture. In line with this, others have reported that the implementation of interdisciplinary huddles increased incident reports and hospital LOS [14]. They concluded that while the intervention could improve care delivery in complex cases, the manner of implementation was paramount, as suboptimal implementation might have a negative impact on performance. Evaluation of the implementation of electronic systems, as assessed by qualitative semi-structured interviews, concluded that it took time to get acquainted with the new system and new routines, which disrupt work efficiency and ultimately impact hospital LOS.

**Table 3. Mortality and readmission rates.**

|  | Intervention (n = 2786) | Control (n = 2610) | HR (95% CI) | p-value* |
|---|---|---|---|---|
| 30-day mortality, n (%) | 76 (2.8%) | 75 (2.9%) | 0.90 (0.65–1.24) | 0.517 |
| 30-day readmission, n (%) | 202 (7.4%) | 166 (6.5%) | 1.14 (0.93–1.40) | 0.204 |
| 1-year mortality, n (%) | 310 (11.3%) | 283 (11.0%) | 0.97 (0.83–1.14) | 0.747 |
| 1-year readmission, n (%) | 355 (12.9%) | 360 (14.0%) | 0.92 (0.80–1.07) | 0.270 |

*All analyses adjusted for age, sex, Charlson comorbidity score, cause of index admission, and admission period (2019–2020 vs. 2021–2022).

We can only speculate what the underlying causes for the results in our study were, but several challenges related to the implementation of new healthcare measures could have contributed. If healthcare staff were not adequately trained and experienced with the PSM, incorporating its use into daily practice and adapting to new routines could have been time-consuming and disruptive to workflow, a known challenge associated with the implementation of new technology in healthcare settings. [11,13]. Furthermore, the identification of more risks or clinical issues that otherwise would have been overlooked could have led to additional evaluation and treatment, ultimately postponing hospital discharge [15]. Despite including control bays where PSMs were not implemented as a sensitivity analysis in our study, we cannot exclude that temporal trends in patient populations could have diluted the effects of PSMs and influenced the results of the study. Particularly, the COVID-19 pandemic could have biased our results, as Norway experienced distinct waves of new cases in early spring 2020 (control period), and spring 2021 and 2022 (intervention period). While baseline patient characteristics were similar between periods, admittance patterns were nevertheless influenced by the pandemic, with fewer elective admissions. Other contextual shifts, such as increased staffing during the intervention period, may also have played a role. However, as these changes would be expected to affect all bays equally, and no comparable worsening of outcomes was observed in sensitivity analyses of control bays, they are unlikely to explain the observed patterns.

## Strengths and limitations

Due to the observational nature of the study design, we cannot exclude that confounding factors influenced the measured endpoints. We did, however, include a control arm with orthopedic bays that did not implement PSMs in the time period, in contrast to most previous studies evaluating the effect of patient safety interventions [11]. Furthermore, we found no differences in baseline characteristics before and after PSM implementation, which strengthens the validity of our results. However, since some outcomes had seemingly similar rates but differences from the regression yielded $p < 0.05$, there may be small imbalances in covariates that influenced the results. The study lacks qualitative assessments, such as staff perceptions or usability evaluations, and quantitative measures such as usage logs or other indicators of user engagement, which could have provided insights into the cause for the apparent limited effect of the PSM implementation. While informal diffusion of PSM-related practices to bays without PSM implementation during the later time period cannot be fully excluded, sensitivity analyses did not show significant changes in LOS in these control bays, suggesting that diffusion is unlikely to explain the findings. The study was conducted in the orthopedic bays of a single hospital, which limits the generalizability of the findings to other departments, hospitals, or healthcare systems, particularly those with different organizational structures, patient populations, or digital infrastructure. This study relied on routinely collected EHR data that were not originally intended to answer the specific research question, which may introduce risks of misclassification bias and unmeasured confounding. In particular, concurrent safety initiatives or policy changes at the hospital level could not be fully accounted for. However, a comprehensive internal data extraction monitoring protocol was implemented, including code review, validation of extracted variables against random manual samples, and multidisciplinary consensus meetings, to mitigate these risks (S1 Appendix). Finally, this evaluation was restricted to factors that were visualized on the PSMs, so although infection rates were reported in other studies, this was not part of the scope of the present study.

## Conclusion

In this study, we did not observe a reduction in hospital LOS following the implementation of PSMs. Additionally, a decrease in nutritional screening and fall risk assessments was seen post-implementation of PSMs. While there was an increase in medication reconciliation and NEWS2 scoring during hospitalization subsequent to the implementation of PSMs, the overall effectiveness of the intervention remains limited. Future research should aim to validate our findings and incorporate qualitative assessments to identify barriers and facilitators that may influence the efficient implementation and utilization of PSMs.

## Supporting information

**S1 Table. Baseline characteristics for bays where patient safety monitors were not implemented.** *p-value for difference between intervention and control group, t-test for age, Chi-square test otherwise.
(DOCX)

**S2 Table. Odds ratio (OR) for safety endpoint in the period after implementation of patient safety monitors (PSM) compared to before for intervention wards and control wards (where PSM was not implemented).** All analyses adjusted for age, sex, Charlson comorbidity score, cause of index admission, and admission period (2019–2020 vs. 2021–2022).
(DOCX)

**S3 Table. Number of NEWS scores taken during the index hospital stay for the intervention wards and control wards (where PSM was not implemented).** All analyses adjusted for age, sex, Charlson comorbidity score, cause of index admission, and admission period (2019–2020 vs. 2021–2022).
(DOCX)

**S4 Table. Time-to-event analyses.** All analyses adjusted for age, sex, Charlson comorbidity score, cause of index admission, and admission period (2019–2020 vs. 2021–2022).
(DOCX)

**S1 Appendix. Predefined study protocol.**
(PDF)

**S2 Appendix. Predefined statistical analysis plan.**
(PDF)

**S3 Appendix. Signed protocol for internal monitoring of data extraction in the study.**
(PDF)

## Author contributions

**Conceptualization:** Rune Bruhn Jakobsen, Johanna A. Gjestland, Magnus N. Lyngbakken, Inge Skråmm, Helge Røsjø.

**Data curation:** Kristian Berge, Torbjørn Wisløff, Kristine Lippestad, Olav Lenvik, Haldor Husby.

**Formal analysis:** Torbjørn Wisløff, Kristine Lippestad, Olav Lenvik, Haldor Husby.

**Investigation:** Magnus N. Lyngbakken, Helge Røsjø.

**Methodology:** Kristian Berge, Torbjørn Wisløff, Magnus N. Lyngbakken, Helge Røsjø.

**Project administration:** Magnus N. Lyngbakken, Helge Røsjø.

**Resources:** Helge Røsjø.

**Supervision:** Magnus N. Lyngbakken, Helge Røsjø.

**Writing – original draft:** Kristian Berge, Torbjørn Wisløff, Kristine Lippestad.

**Writing – review & editing:** Kristian Berge, Torbjørn Wisløff, Kristine Lippestad, Rune Bruhn Jakobsen, Johanna A. Gjestland, Magnus N. Lyngbakken, Inge Skråmm, Helge Røsjø.

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
