## [Decision Letter · Decision Letter 0]

21 Jul 2025

Dear Dr. Berge,

Thank you for submitting your manuscript to PLOS ONE. After careful consideration, we feel that it has merit but does not fully meet PLOS ONE’s publication criteria as it currently stands. Therefore, we invite you to submit a revised version of the manuscript that addresses the points raised during the review process.

We look forward to receiving your revised manuscript.

Kind regards,

Teerapon Dhippayom

Academic Editor

PLOS ONE

Journal Requirements: 

2. In the online submission form, you indicated that [Data will be make available upon reasonable request.].

[KB has received speaker honoraria from Boehringer Ingelheim, Novartis, Amgen, and AstraZeneca.].   

We note that one or more of the authors are employed by a commercial company: name of commercial company.

Within your Competing Interests Statement, please confirm that this commercial affiliation does not alter your adherence to all PLOS ONE policies on sharing data and materials by including the following statement: ""This does not alter our adherence to  PLOS ONE policies on sharing data and materials.” (as detailed online in our guide for authors http://journals.plos.org/plosone/s/competing-interests) . If this adherence statement is not accurate and  there are restrictions on sharing of data and/or materials, please state these. Please note that we cannot proceed with consideration of your article until this information has been declared.

Additional Editor Comments:

Thank you for submitting this manuscript evaluating the impact of PSM on patient outcomes. The topic is important and relevant to efforts aimed at improving hospital safety. After reviewing the manuscript, I would like to raise the following points:

#1 Study Design and Protocol Registration: The study is described as a “pragmatic, pseudo-randomized trial,” but the allocation method suggests that the design is more accurately classified as a quasi-experimental study. Please consider revising this terminology to reflect the actual methodological approach. Additionally, please clarify whether the study protocol was prospectively registered. If not, this should be acknowledged as a limitation due to potential reporting bias.

#2 Clinical Implications and Subgroup Effects: Although the overall findings did not demonstrate statistically significant benefits from the PSM intervention, it remains possible that specific patient subgroups, such as those at higher risk for delirium or falls, might benefit more clearly. I encourage the authors to consider performing subgroup analyses to explore PSM potential effects of certain groups.

#3 Interpretation of Findings and Cost-Effectiveness Recommendation: The manuscript suggests that cost-effectiveness analysis (CEA) of the PSM intervention may be warranted. Given that the intervention did not show meaningful clinical benefit in the overall analysis, this recommendation appears premature. Please reconsider this suggestion and revise it to align more appropriately with the findings.

Reviewers' comments:

Reviewer's Responses to Questions

**Comments to the Author**

1. Is the manuscript technically sound, and do the data support the conclusions?

Reviewer #1: Yes

Reviewer #2: Yes

2. Has the statistical analysis been performed appropriately and rigorously?

Reviewer #1: Yes

Reviewer #2: Yes

3. Have the authors made all data underlying the findings in their manuscript fully available?

Reviewer #1: Yes

Reviewer #2: Yes

4. Is the manuscript presented in an intelligible fashion and written in standard English?

Reviewer #1: Yes

Reviewer #2: Yes

Reviewer #1: A well written paper that evaluates the implementation of a Patient Safety Monitor on clinical outcomes in a pre-/post-study. Most outcomes do not change, but some do actually get worse. Impressively the study is conducted through COVID with few changes in baseline characteristics.

There are only a few minor revisions that I would suggest for the benefit of the understanding of international readers.

1. Could you kindly add a picture of the monitor: the design of the interface might have significant impact on usability and clinical translation of warning signals.

2. I am not sure whether you mean 'bay' - a bay has usually got 4-6 beds, a ward has 20-30 beds. I would struggle to understand why units of 6 bays would have autonomy to implement or not implement? Why would this be?

3. I don't think that this is a pseudo-randomised trial - it is a pre-post study with a parallel group?

4. I would value the view of the authors why some of the outcomes might have worsened. What other things happened at the time? Did teams get complacent given an over-reliance on new technology? Can you comment on bed occupancy and number of operations per year that might have been affected by COVID?

5. I have one other question: why did you use length of stay as your main outcome for a safety intervention? Death, ICU admission might be more appropriate?

Reviewer #2: • This study presents a well-conducted, pragmatic pseudo-randomized trial evaluating the impact of electronic Patient Safety Monitors (PSMs) in a real-world hospital setting. Despite finding no significant reduction in hospital length of stay or key patient safety outcomes, the research offers valuable insights into the practical challenges of implementing PSMs. This work contributes meaningfully to the evidence base and highlights the need for further research into the effective integration of PSMs. Overall, this is a well-executed and timely study that offers valuable real-world insights into the limited effectiveness of digital patient safety interventions in routine clinical practice.

• The manuscript would benefit from a brief justification for selecting orthopaedic bays as the setting for PSM implementation, as this could help readers assess the generalizability and contextual applicability of the findings. I am thinking several other clinical settings could be suitable—perhaps even more suitable—for evaluating the effectiveness of PMSs, especially if the goal is to assess broad patient safety outcomes, clinical workflow improvements, or preventable harms.

• The difference in LOS is statistically significant (p=0.046) but clinically trivial. The abstract and key messages say “no effect” on LOS, which might appear contradictory unless clarified as not clinically meaningful. The discussion restates that LOS increased slightly and was statistically significant, but does not highlight how small the actual difference was.

• They speculate about implementation challenges (training, workflow disruption), but no data on user engagement, usage logs, or staff feedback is presented or acknowledged as a gap.

• The risk of temporal confounding, especially given that the study spans the COVID-19 pandemic period, which could have independently influenced hospital workflows, admissions, and outcomes. You can be 1-2 sentences describing these issues!!

• The discussion is well-structured and places the study’s findings within the broader context of existing literature and global patient safety initiatives. The limitations are clearly presented and helpfully explain potential sources of bias, reflecting strong awareness of the study’s methodological constraints and enhancing its overall credibility. However, findings may not be generalizable to other healthcare settings with different organizational structures, patient populations, or digital infrastructure. Please considered to be as one of the limitations.

• Although important variables such as age, sex, Charlson Comorbidity Index, and cause of admission were adjusted for, other potentially relevant confounders—such as staffing levels, nurse-to-patient ratios, concurrent safety initiatives, or recent policy changes—may not have been fully accounted for. Why?

• Staff rotating between intervention and control bays or informal diffusion of PSM practices to non-PSM areas could dilute the observed effects, but this possibility is not well addressed.

**Do you want your identity to be public for this peer review?** For information about this choice, including consent withdrawal, please see our Privacy Policy

Reviewer #1: **Yes: ** Christian P Subbe

Reviewer #2: No

---

## [Author Response · Author response to Decision Letter 1]

10 Sep 2025

As per email instructions, a separate file (Response to Reviewers) has been uploaded with responses to reviewer and editor comments.

---

## [Decision Letter · Decision Letter 1]

6 Oct 2025

Pragmatic, quasi-experimental, pseudo-randomized clinical trial to assess the impact of patient safety monitors on clinical and patient safety outcomes: the Akershus Clinical Trial (ACT) 1

PONE-D-25-28885R1

Dear Dr. Berge,

We’re pleased to inform you that your manuscript has been judged scientifically suitable for publication and will be formally accepted for publication once it meets all outstanding technical requirements.

Kind regards,

Teerapon Dhippayom

Academic Editor

PLOS ONE

Additional Editor Comments (optional):

Thank you for submitting the revised manuscript. I have carefully reviewed the responses to my previous comments, as well as those raised by both reviewers. I am satisfied that all concerns have been adequately addressed, and the revised version demonstrates clear improvement. The manuscript now meets the journal’s standards for publication.

Reviewers' comments:

Reviewer's Responses to Questions

**Comments to the Author**

Reviewer #1: All comments have been addressed

Reviewer #2: All comments have been addressed

2. Is the manuscript technically sound, and do the data support the conclusions?

Reviewer #1: Yes

Reviewer #2: Yes

3. Has the statistical analysis been performed appropriately and rigorously?

Reviewer #1: Yes

Reviewer #2: Yes

4. Have the authors made all data underlying the findings in their manuscript fully available?

Reviewer #1: No

Reviewer #2: Yes

5. Is the manuscript presented in an intelligible fashion and written in standard English?

Reviewer #1: Yes

Reviewer #2: Yes

Reviewer #1: I am satisfied with the replies by the authors. I believe that the paper adds to the body of evidence and that the limitation in relation to publication of complete data is likely to be within the legal framework of the authors.

Reviewer #2: (No Response)

**Do you want your identity to be public for this peer review?** For information about this choice, including consent withdrawal, please see our Privacy Policy

Reviewer #1: **Yes: ** Dr Christian Peter Subbe

Reviewer #2: **Yes: ** Sajesh K Veettil

---

## [Editor Report · Acceptance letter]

PONE-D-25-28885R1

PLOS ONE

Dear Dr. Berge,

I'm pleased to inform you that your manuscript has been deemed suitable for publication in PLOS ONE. Congratulations! Your manuscript is now being handed over to our production team.

Kind regards,

on behalf of

Dr. Teerapon Dhippayom

Academic Editor

PLOS ONE